# Physiotherapeutic Methods in the Treatment of Cervical Discopathy and Degenerative Cervical Myelopathy: A Prospective Study

**DOI:** 10.3390/life12040513

**Published:** 2022-03-31

**Authors:** Grzegorz Mańko, Małgorzata Jekiełek, Tadeusz Ambroży, Łukasz Rydzik, Jarosław Jaszczur-Nowicki

**Affiliations:** 1Department of Biomechanics and Kinesiology, Institute of Physiotherapy, Jagiellonian University Collegium Medicum, 31-126 Kraków, Poland; g.manko@uj.edu.pl; 2ORNR “Krzeszowice”, Rehabilitation Center, Daszyńskiego 1, 32-065 Krzeszowice, Poland; 3Institute of Physiotherapy, Faculty of Health Sciences, Jagiellonian University Collegium Medicum, 31-008 Kraków, Poland; malgorzata.jekielek@gmail.com; 4Institute of Sports Sciences, University of Physical Education, 31-571 Kraków, Poland; tadek@ambrozy.pl; 5Department of Tourism, Recreation and Ecology, University of Warmia and Mazury, 10-719 Olsztyn, Poland

**Keywords:** degenerative cervical myelopathy, cervical discopathy, exercise interventions, functional recovery, chronic disease, physical activity

## Abstract

Spinal dysfunctions are very common in the population. However, there is still a lack of information on how to diagnose and treat them properly. The common causes of spinal dysfunctions are cervical discopathy and degenerative cervical myelopathy. The aims of the study are to examine whether a combination of manual therapy and stabilometer platform exercises can be effective in treating cervical discopathy and degenerative cervical myelopathy, and the possibility of observing the differences between patients suffering from the above diseases. The study involved 40 patients referred for rehabilitation, who formed two groups of 20 people. The first group consisted of patients suffering from cervical discopathy, the second group consisted of patients affected by degenerative cervical myelopathy. During therapy, manual therapy techniques and a stabilometric platform were used. The Neck Disability Index and Pain Numeric Rating Scale were used for clinical evaluation. The correlation between the existing diseases and the results obtained in the Neck Disability Index and Pain Numeric Rating Scale was examined. The distribution of patient responses in questions of the Neck Disability Index was also checked. Clinical evaluation was performed twice, before the start of therapy and after a two-week rehabilitation treatment. The study showed a significant difference between the patients’ results before the start of therapy and after the end of the rehabilitation stay in both used questionnaires (*p* = 0.00). A difference in the distribution of responses between the two groups after therapy was also found in the Neck Disability Index (*p* = 0.018) and in the Pain Numeric Rating Scale (*p* = 0.043). The study shows that manual therapy and exercises using the stabilometric platform are effective methods of treating both patients with cervical discopathy and patients with degenerative cervical myelopathy. It was also noted that, when comparing groups of patients, patients with degenerative cervical myelopathy tend to have greater disturbances in concentration-related activities, such as reading, focusing, driving, sleeping, and resting.

## 1. Introduction

The spine is a diagnostic and therapeutic challenge [1,2,3]. Although new research is emerging, there are still many unknowns to discover. Degenerative changes of the spinal column are related with age and are very common [4,5]. They are often diseases leading to lower quality of life and increased occurrence of disability [6,7,8]. One of the most common spine dysfunctions is degenerative cervical myelopathy [2,8,9,10,11,12]. This is a condition in which the spinal canal narrows, which causes increased pressure on the spinal cord [13]. The concept of cervical myelopathy can be defined as nerve implantation, injury, or disease of the spinal cord or the vertebrae of the neck [14,15]. According to the literature, cervical degenerative myelopathy (DCM) results from cervical compression of the spinal cord as a result of age-related changes in the cervical spine and affects up to 2% of adults, leading to progressive disability [16,17]. The causes of degenerative cervical myelopathy may be static overload and loss of disc height, which leads to incorrect biomechanics of the spine. Another cause may be overload by incorrect, repeated movements. Continuous irritation of the spinal cord leads to initiation of an inflammatory reaction and vascular changes, which may result in ischemia and loss of neuronal cells [18]. Patients complain about neck stiffness, paresis, and dizziness, as well as sleep disorders and impaired cognitive functions [13,19]. Importantly, pain does not always occur [13]. Currently, degeneration of the intervertebral disc is a common disease of the spine, with changes related with age [19]. This condition is associated with dehydration, which reduces the disc’s height, reduces elasticity, and changes the biomechanics of the discs [20]. In the late period of degeneration, herniation of the nucleus may occur, pressing on the spinal canal and the spinal nerves, which can cause pain, sensory disturbances, and paresis in the upper limbs [21]. Herniation of the intervertebral disc, besides mechanical effects, such as pressure on nerves and vessels, can cause an inflammatory reaction [22]. A spinal disk herniation compressing the spinal cord may cause degenerative cervical myelopathy [13]. Compression of the blood vessels by herniation may lead to headaches, dizziness, or loss of consciousness [23]. These are the symptoms typical of myelopathy. Cervical discopathy is damage to the intervertebral disc in the cervical segment, which may lead to the appearance of a hernia [24,25]. This condition is caused when disc herniation or other pathologies of the disc and cervical spine occur. Symptoms of discopathy include inflammatory reactions in the area of the spinal nerve, causing pain or local swelling [26,27].

The aim of the study was to examine the influence of a combination of manual therapy and exercises on a stabilometric platform as a method of treating cervical discopathy and degenerative cervical myelopathy, and to observe possible differences in terms of pain assessment and the level of disability of the cervical spine.

## 2. Materials and Methods

The study was a prospective study. The representativeness of the subjects was provided by random selection of patients for treatment groups. Patients in the research groups were randomized using a randomizing program from among those admitted to rehabilitation based on a referral. The first group, from which 20 patients were drawn, was the group where patients were diagnosed as suffering from cervical discopathy, based on magnetic resonance imaging diagnostics in these people, which determined spinal cord modeling, spinal cord compression, and intervertebral disc herniation. The second group were patients who received a diagnosis of degenerative cervical myelopathy, which also included compression of the spinal cord and intervertebral disc herniation; however, this group were without modeling of the spinal cord. A total of 20 people were also selected from this group using a randomizing program. The study was conducted with the consent of the patients, guaranteeing anonymity. Patients were informed about the research technique and their purpose. Questionnaires of patients participating in the study were coded with the letters D and M (discopathy and myelopathy, respectively), the patient’s initials, and a number from 1 to 20, to make it impossible to identify the examined person. The study design received a positive assessment from the Commission for Ethics in Scientific Research at the University of Warmia and Mazury in Olsztyn (approval number 9/2018). The study design is presented in Figure 1.

### 2.1. Participants

The study included a selected group of patients undergoing a two-week rehabilitation stay at the Krzeszowice Movement Rehabilitation Center, aged 60–74 years. In the group suffering from cervical discopathy (D), there were 12 women and 8 men, while in the group with degenerative cervical myelopathy (M), there were 14 women and 6 men. The median age in the cervical discopathy (D) group was 66 years (IQR = 8.5), while the median in the degenerative cervical myelopathy (M) group was 66.5 years (IQR = 8.5).

*Criteria for inclusion in the study:* Expressing written consent to participate in the study, i.e., submission to manual therapy and exercises on the stabilometric platform; expressing written consent to assess the patient’s condition before and after therapy using the Pain Numeric Rating Scale and Neck Disability Index; physical condition enabling movement and self-service in everyday activities, as assessed by the physician referring the patient to the rehabilitation stay; mental state, ensuring cooperation during the performance of tests, as assessed by a physician referring to a rehabilitation stay; medical referral for physiotherapy.

*Exclusion criteria for the test:* Cancer; schizophrenia; dementia; Parkinson’s disease; Alzheimer’s disease; stroke; endoprosthetic surgery; neurological diseases; multiple sclerosis; amyotrophic lateral sclerosis; muscular dystrophy; taking analgesic medications; diabetes; unstable coronary artery disease; unstable blood pressure; acute period of cardiological and rheumatic diseases; inflammatory bowel disease; injuries to the cervical spine in the past.

### 2.2. Instruments

All 40 patients qualified for the study and were undergoing therapy. They were examined using the Neck Disability Index, containing 10 questions about everyday life, addressing pain intensity, care, lifting objects, work, reading, headache, concentration, driving, sleep, and rest. For each of these elements, patients were presented with a scale from 0 to 5 to assess their condition. As a result of the test, a maximum of 50 points could be attained. A score of 0–4 points indicated no disability, a score of 5–14 indicated mild disability, a score of 15–24 indicated moderate disability, a score of 25–34 indicated severe disability, and a score of 35–50 indicated complete disability. The Pain Numeric Rating Scale was also used, in which patients scored, on a scale of 1–10 points, the intensity of pain usually felt during the last week, and the minimum and maximum pain level during the last week. The obtained average of the above 3 results allowed us to obtain the average pain of the patient during the week.

### 2.3. Interventions

In both groups, manual therapy techniques including deep tissue massage of the cervical spine and shoulder girdle were used. Mobilization of the cervical spine with traction, according to the concept of *Orthopedic Manual Therapy* by Kaltenborn and Evjenth, was also used, which was conducted by qualified physiotherapy specialists (physiotherapists conducting the exercises have completed training in a given method of therapy). The Zebris stabilometric platform was also used, along with the default training plan available within the platform, in the form of an interactive game. The game consisted of balancing the body in the frontal plane in order to catch tomatoes falling from a table into a basket. Deep tissue massage therapy lasted 20 min, and the training session on the stabilometric platform lasted 10 min. The procedures were performed from Monday to Friday during a two-week rehabilitation period. Patient clinical control was performed using the Neck Disability Index and Pain Numeric Rating Scale before the start of therapy and after the rehabilitation period.

### 2.4. Data Analysis

In order to describe the variables occurring during the analysis of the study, the central tendency indicator median was used, in addition to the measure of dispersion, which was the interquartile range. In order to examine the existence of correlations between the studied variables, the Spearman correlation test was used. The Wilcoxon test was also used to test the significance of differences obtained before and after physiotherapy in groups D and M, and the Mann–Whitney U test was used to check the significance of the difference between the results obtained before and after therapy in both groups studied.

## 3. Results

Before starting the studies, in the Neck Disability Index questionnaire, patients in the group affected by cervical discopathy obtained results with a median of 24 points, a minimum of 9 points, and a maximum of 36 points. In the group of patients suffering from degenerative cervical myelopathy, these results were 35, 16, and 45, respectively. A detailed distribution of the data is presented in Table 1.

A statistically significant difference was found in the Mann–Whitney U test between the mean results of the Neck Disability Index in patients suffering from cervical discopathy and patients suffering from degenerative cervical myelopathy (*p* = 0.00).

Before therapy in group D, 25% of patients in the Neck Disability Index questionnaire obtained a result allowing them to be classified into the group with mild disability, 30% of patients were assigned to the moderate group, 40% were assigned to the severe group, and 5% of patients were assigned to the group with complete disability. There were no patients with no or mild disability in the group affected by degenerative cervical myelopathy. A proportion of 20% of patients were described as people with moderate disability, 30% were described as people with severe disability, and half as many patients were described as people with total disability. Table 2 shows the distribution of patients in groups D and M in individual disability groups.

The study examined the level of pain in patients in both groups before starting therapy. In group D, the median result was 4.5 points, the minimum and maximum were 3 and 6 points, respectively; meanwhile, in group M, the median was 4.67 points, the minimum was 3 points, and the maximum was 8.67 points. Table 1 shows the distribution of results on the PNRS scale in patients of both groups.

After completion of therapy, the median points obtained in the Neck Disability Index for patients in group D dropped to 4.5 points. The minimum value obtained in this group was 0 and the maximum was 9 points. In the group affected by degenerative cervical myelopathy, the median was 10 points, while the minimum and maximum were 1 and 18 points, respectively. Table 3 shows the distribution of points obtained by patients in NDI in groups D and M.

In the group affected by discopathy in the cervical spine, half of the patients were classified into the group without disability, and the other half were classified into the group of patients with mild disability. In the group suffering from cervical myelopathy, 20% of patients showed no disability, 60% of respondents showed mild disability, and 20% of patients were classified as moderately disabled. A detailed distribution of the data is presented in Table 4.

After the therapy, the severity of pain in patients in groups D and M was measured again. In the group of patients suffering from cervical discopathy, the median of the obtained points was 2 points, with a minimum of 1 point and a maximum of 3.33 points. In the cervical myelopathy group, these values were 2.33, 1.33, and 3.67 points, respectively. Table 3 presents the distribution of the number of points obtained by patients in both groups on the PNRS scale after treatment.

After treatment in patients in group D, the median decrease in points obtained in the Neck Disability Index was 20 points. The smallest decrease was 11 points and the largest was 31 points in relation to the initial result. In the group M, the median decrease was 25.5 points, while the minimum and maximum decreases were 14 and 32 points, respectively. After the end of therapy in group D, the number of points on the PNRS scale decreased by 2.67 points compared with the initial value. The minimum decrease was 1.67 points, and the maximum decrease was 3.33 points. For group M, the median decrease was 2.33 points, while the minimum and maximum decreases were 1.33 and 5.67 points, respectively. Table 5 shows the difference between the points obtained before and after therapy in the subjects.

After the end of therapy in group D, the number of points on the PNRS scale decreased by 2.67 points compared with the initial value. The minimum decrease was 1.67 points, and the maximum decrease was 3.33 points. For group M, the median decrease was 2.33 points, while the minimum and maximum decreases were 1.33 and 5.67 points, respectively. Table 5 shows the difference in points obtained by patients on the Pain Numeric Rating Scale before and after therapy.

A Mann–Whitney U test was also performed to compare the average improvement in the results obtained by the subjects in both groups using the Neck Disability Index and the Pain Numeric Rating Scale. A statistically significant decrease was obtained in the number of points obtained in NDI (*p* = 0.018), and a statistically significant decrease was obtained in the number of points obtained in PNRS (*p* = 0.043).

The significance of improving the results of patients before and after therapy in individual groups D and M was also checked using the Wilcoxon test, where a statistically significant difference was found both in the Neck Disability Index and the Pain Numeric Rating Scale, obtaining *p* = 0.00 in both cases.

The statistical analysis also examined the correlation between the disease entity occurring and the decrease in the number of points in the Neck Disability Index and the PNRS pain scale. There was a statistically significant moderate correlation (correlation coefficient = 0.376) between the disease entity present and a decrease in the number of points in the Neck Disability Index spreadsheet (*p* = 0.017). There was no statistically significant correlation between disease and decrease in the number of points in the Pain Numeric Rating Scale (*p* = 0.053, correlation coefficient = −0.309).

During the study, the scoring of responses in the Neck Disability Index in individual disease entities was also analyzed. It has been reported that, in activities such as reading, concentration, driving, sleep, and rest, patients with degenerative cervical myelopathy statistically indicated higher-scoring responses, thus determining a higher degree of discomfort than patients affected by cervical discopathy. The same is true with neck pain, where patients suffering from degenerative cervical myelopathy indicated higher scores than patients affected by cervical discopathy. Table 6 shows the difference in the answers given by patients before starting therapy.

## 4. Discussion

The aim of this study was to test the effectiveness of combined manual therapy and exercises on a stabilometric platform in the treatment of cervical discopathy and degenerative cervical myelopathy, and the possibilities of the Neck Disability Index in differentiating between the diseases. Results of this study showed that manual therapy and training on a platform reduced the level of disability among patients with cervical discopathy and degenerative cervical myelopathy.

Cervical discopathy and degenerative cervical myelopathy are associated with impairment of physical and psychosocial functions [28,29]. Chronic cervical spine pains lead to muscle weakness and a decrease in quality of life. The Neck Disability Index was found to be a good method of distinguishing patients with different levels of disability amongst patients with chronic neck pain [30]. Patients reported poor sleep quality [31]. One of the biggest problems reported by patients with cervical discopathy and myelopathy is pain. Pain leads to a reduction in the activity of patients and a decrease in their quality of life [31,32]. Chronic pain contributes to depression [32]. Patients with psychological disease respond less well to treatment and obtain poorer outcomes compared with nondepressed patients [33].

It should be remembered that discopathy and myelopathy are different concepts. Not every diagnosed discopathy gives symptoms of myelopathy. Advanced stage of disc herniation or bulging can lead to symptoms of myelopathy [8]. Myelopathy and discopathy also limit the activities of daily living (ADL), housework, work, and driving [34]. The research in our study showed that patients with cervical myelopathy have greater cognitive problems, poorer sleep quality, and problems with social functioning, performance of tasks, and driving a car. According to the literature, women suffer from chronic neck pain more often than men [35]. Therefore, more women took part in our research. Among patients with discopathy, kinesiophobia often occurs [8,26]. It is an irrational fear of movement in the interest of preventing pain or trauma [26]. The level of cervical discopathy can give different symptoms. A lower cervical discopathy contributes to pain in the area of the scapula with or without radiation to the upper limb [20,36]. Degenerative changes in the upper cervical spine may induce neuralgia. Proliferative changes in the uncovertebral joints (located on each side of cervical discs between levels C3 and C7) may lead to compression of the vertebral artery [36]. Some research suggests that the severity of degenerative changes in the intervertebral joints and intervertebral discs visible on X-rays are not always associated with pain levels and disability [35].

Patients with degenerative cervical myelopathy score worse for cognitive function according to the Neck Disability Index than patients with cervical discopathy, and the results of this study may be a guideline for the differentiation between cervical discopathy and degenerative cervical myelopathy. However, there is still a need for new research to differentiate between these two diseases. The literature focuses mainly on the surgical treatment of spine diseases. More research is needed into the possibilities of physiotherapeutic treatment.

Study limitations: The number of participants in the present study was limited. Future research should investigate a larger population. Additionally, the examined patients were under the care of physiotherapy for only two weeks of the rehabilitation stay. It would be worthwhile to examine patients who have undergone a longer treatment period. It would also be worth considering the blindness of the therapists participating in the therapy. The present study has a very broad perspective because an increasing number of people are suffering from cervical spine disorders. The subject of the research undertaken requires further research.

## 5. Conclusions

The present study shows that manual therapy and exercises using a stabilometric platform are effective methods for treating both patients with cervical discopathy and patients with degenerative cervical myelopathy. It has been reported that patients suffering from degenerative cervical myelopathy achieve higher scores in the Neck Disability Index than patients with cervical discopathy, which indicates that they subjectively assess their disability higher than patients with cervical discopathy. It was also noted that patients with degenerative cervical myelopathy were more impaired in terms of concentration-related activities, such as reading, focusing, driving, sleeping, and relaxing. In terms of pain intensity and daily activities, such as care and lifting, patients with cervical discopathy or degenerative cervical myelopathy show a similar level of discomfort associated with the diseases.

### Clinical Implication

Considering that patients with degenerative cervical myelopathy were more impaired in terms of concentration-related activities, attention should be paid to the above elements during the interview and examination of patients reporting to a physiotherapy clinic with disorders of the cervical spine. These elements may become indicators in the screening of patients with such disorders before the use of diagnostic tools using magnetic resonance imaging.

## Figures and Tables

**Figure 1 life-12-00513-f001:**
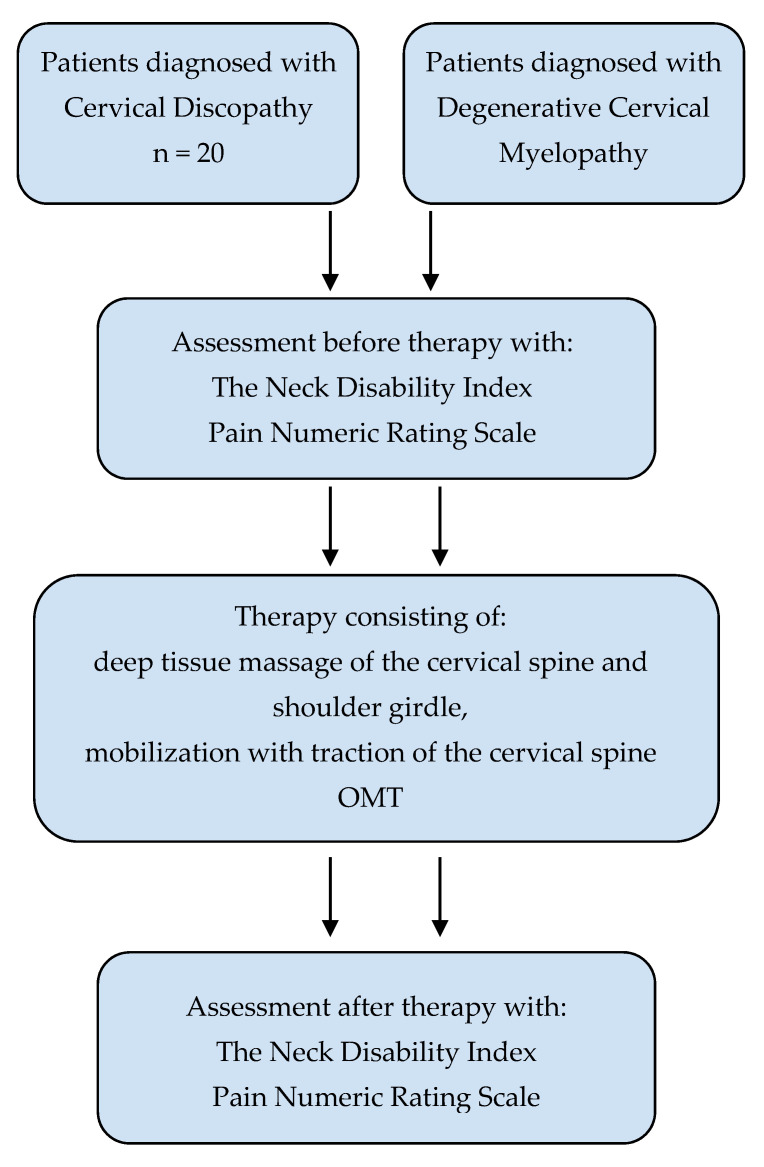
The study design.

**Table 1 life-12-00513-t001:** Distribution of points obtained in the Neck Disability Index and the Pain Numeric Rating Scale in patients in group D and in group M before therapy.

Points	Group D				Group M			
	Median	IQR	Min	Max	Median	IQR	Min	Max
NDI	24	16.25	9	36	35	13	16	45
PNRS	4.5	1.25	3	6	4.67	1.58	3	8.67

NDI—Neck Disability Index; PNRS—Pain Numeric Rating Scale; IQR—interquartile range; Min—minimum; Max—maximum.

**Table 2 life-12-00513-t002:** Distribution of patients in groups D and M in individual disability groups before applying the therapy.

	Group D		Group M	
Disability	n	%	n	%
No disability	0	0	0	0
Mild disability	5	25	0	0
Moderate disability	6	30	4	20
Severe disability	8	40	6	30
Complete disability	1	5	10	50

**Table 3 life-12-00513-t003:** Number of points obtained in the Neck Disability Index and the Pain Numeric Rating Scale in patients in group D and in group M after therapy.

Points	Group D				Group M			
	Median	IQR	Min	Max	Median	IQR	Min	Max
NDI	4.5	5	0	9	10	17	1	18
PNRS	2	0.67	1	3.33	2.33	1.25	1.33	3.67

NDI—Neck Disability Index; PNRS—Pain Numeric Rating Scale; IQR—interquartile range; Min—minimum; Max—maximum.

**Table 4 life-12-00513-t004:** Shows the distribution of patients in groups depending on the level of disability after using the therapy.

Distribution of Patients in Groups D and M in Individual Disability Groups after Completing Therapy	Group D		Group M	
Disability	n	%	n	%
No disability	10	50	4	20
Mild disability	10	50	12	60
Moderate disability	0	0	4	20

**Table 5 life-12-00513-t005:** Difference of points obtained in the Neck Disability Index before and after therapy.

Points	Group D				Group M			
	Median	IQR	Min	Max	Median	IQR	Min	Max
NDI	20	11	9	31	25.5	9	14	32
PNRS	2.67	0.92	1.67	3.33	2.33	0.92	1.33	5.67

NDI—Neck Disability Index; PNRS—Pain Numeric Rating Scale; IQR—interquartile range; Min—minimum; Max—maximum.

**Table 6 life-12-00513-t006:** Number of points obtained in the Neck Disability Index in individual questions in patients in group D and in group M before starting therapy.

	Group D				Group M			
Question	Median	IQR	Min	Max	Median	IQR	Min	Max
1	3	1.75	1	4	3	2	2	4
2	3	2	1	4	3	2	2	4
3	4	2	1	4	3	2	1	4
4	1	1	0	4	4	2	1	5
5	2	2	1	4	4	2	1	5
6	1	2	0	4	4	1	1	5
7	2	2	1	4	3.5	1	1	5
8	2.5	3	1	4	4	1	1	5
9	2.5	2	0	4	3	2.5	0	4
10	2	2	1	4	4	2.5	1	5

## Data Availability

All data are presented in the study.

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
