# Peer review of "Physiotherapeutic Methods in the Treatment of Cervical Discopathy and Degenerative Cervical Myelopathy: A Prospective Study"

_life, 2022, doi:10.3390/life12040513_

Round 1

Reviewer 1 Report

Despite the good results obtained, the sample could be increased to at least 50-60 participating subjects.
The selection criteria could be written on lines separated by commas.
The descriptive data of the sample could be given without the need for tables. Perhaps there is an excess of tables in results.
Some p-values ​​for the most relevant results should be given in the abstract.
From my point of view, there is a low number of references.
The professionals who perform the evaluation and whether they are blinded are not specified.

Author Response

Dear Reviewer,

Thank you very much for your time and valuable comments, which all have been considered and incorporated. The detailed list of responses is given below. We hope that the modifications and explanation will be acceptable for you.

Yours sincerely,

Rydzik, corresponding author

Point 1:Despite the good results obtained, the sample could be increased to at least 50-60 participating subjects.

Answer 1:Thanks a lot for your suggestion. The small sample of patients results from the total number of patients admitted to rehabilitation stays in the center where the research was conducted.

Point 2:The selection criteria could be written on lines separated by commas.

Answer 2:The text has changed

Point 3:The descriptive data of the sample could be given without the need for tables. Perhaps there is an excess of tables in results.

Answer 3:Thank you for your feedback, we have reduced the number of tables.

Point 4:Some p-values ​​for the most relevant results should be given in the abstract.

Answer 4:Thank you, we have added p values to the abstract

Point 5:From my point of view, there is a low number of references.

Answer 5:Thank you for the suggestion, we have improved the text.

Point 6:The professionals who perform the evaluation and whether they are blinded are not specified.

Answer 6:The assessment of patients and physiotherapy was carried out by the same therapists who were assigned to the patient. They cared for patients from the first assessment visit, throughout treatment, and made a final assessment using two questionnaires. They were not blinded because the diagnosis was recorded in the medical records of patients who came to them for therapy.

Reviewer 2 Report

Thank you for permitting me to review 

Please define adequately the objectives of the study 

Please define exactly both groups medical definition of both groups  at the introduction , right now a brief description is in the method section 

Please specifiy "certified " physiotherapist 

Line 66: How can this study be randomized, as it is stated that the aim is to compare 2 types of pathology , this is not a randomized trial this is a prospective study comparing two groups. 

Please delete all one raw tables since they can be replaced with a one line sentence 

Author Response

Dear Reviewer,

Thank you very much for your time and valuable comments, which all have been considered and incorporated. The detailed list of responses is given below. We hope that the modifications and explanation will be acceptable for you.

Yours sincerely,

Rydzik, corresponding author

Point 1:

Please define adequately the objectives of the study

Answer 1:

Thanks for the suggestion, we have improved the section.

Point 2:

Please define exactly both groups medical definition of both groups  at the introduction , right now a brief description is in the method section

Answer 2:

Thank you for the suggestion, we have improved the text.

Point 3:

Please specifiy "certified " physiotherapist

Answer 3:

Thanks for your suggestion, we've completed the section.

Point 4:

Line 66: How can this study be randomized, as it is stated that the aim is to compare 2 types of pathology , this is not a randomized trial this is a prospective study comparing two groups.

Answer 4:

Thank you for your suggestion, we changed the type of research. The study groups were assigned to patients who were randomly assigned to them. From the group of patients with a given disease entity, only a selected number was admitted to the study.

Point 5:

Please delete all one raw tables since they can be replaced with a one line sentence

Answer 5:

Thank you for your feedback, we have reduced the number of tables.

Reviewer 3 Report

The study has a high interest to the readers.

I have some considerations regarding the contents and the methodology.

during the introduction nothing is mentioned about the effectiveness of physiotherapy treatments.

There are several systematic reviews that stratify the rehabilitation approach based on the symptoms and severity of the disease.

I suggest you read this article or similar for implement the introduction: “Management of neck pain and associated disorders: A clinical practice guideline from the Ontario Protocol for Traffic Injury Management (OPTIMa) Collaboration”.

The study should include a control group to justify the effects of the treatment.

The randomization is not clear to me, the groups were set according to the pathology and the patients assigned based on the diagnosis.

Assessment methods lack an assessment of kinesiophobia and an objective evaluation of Cervical Range of Motion.

Please insert a flowchart of the study.

Don't you have a trial registration?

The results are short-term, the use of a follow-up is desirable, there is no blinding and no allocation. In my opinion, these criticalities should be reported in the discussion.

At line 125-127 “The mobilization of the cervical spine according to the concept of Orthopedic Manual Therapy by Kaltenborn and Evjenth was also used, which was conducted by qualified physiotherapy specialists.”.  What kind of mobilizations have been made? Passive? Active-assisted? Have therapeutic exercises been used beyond the stabilometric platform? Please specify that point.

From line 286 to 294: “It has been reported that patients suffering from Degenerative Cervical Myelopathy in Neck Index Disability achieve higher scores than patients with Cervical Discopathy, which indicates that they subjectively assess their disability higher than 288 patients with Cervical Discopathy. It was also noted that patients with Degenerative Cervical Myelopathy had were more impaired in terms of concentration-related activities 290 such as reading, focusing, driving, sleeping and relaxing. In terms of pain intensity and daily activities such as care, lifting, patients with Cervical Discopathy and Degenerative Cervical Myelopathy show a similar level of discomfort associated with the disease. This is important from the perspective of the possibility of differentiating these diseases.”

There are several red flags used in physiotherapy that guide the physiotherapy diagnosis and help the therapeutic action in understanding when a referral is needed.

I think it should be specified that this element can be useful in screening the pathology before the diagnostic finding (MRI). It could be a guide for rehabilitators who have direct access with this type of patient.

Author Response

Dear Reviewer,

Thank you very much for your time and valuable comments, which all have been considered and incorporated. The detailed list of responses is given below. We hope that the modifications and explanation will be acceptable for you.

Yours sincerely,

Rydzik, corresponding author

Point 1:

During the introduction nothing is mentioned about the effectiveness of physiotherapy treatments. There are several systematic reviews that stratify the rehabilitation approach based on the symptoms and severity of the disease. I suggest you read this article or similar for implement the introduction: “Management of neck pain and associated disorders: A clinical practice guideline from the Ontario Protocol for Traffic Injury Management (OPTIMa) Collaboration”.

Answer1:

Thank you for the suggestion, we have improved the text.

Point 2:

The study should include a control group to justify the effects of the treatment.

Answer 2:

Thanks for your suggestion. In our study, the main goal was to check whether the results of the study of groups with various diseases will differ, but we understand that the introduction of a control group would improve the quality of work. We will use the control group in future publications.

Point 3:

The randomization is not clear to me, the groups were set according to the pathology and the patients assigned based on the diagnosis.

Answer 3:

The study groups were assigned to patients who were randomly assigned to them. From the group of patients with a given disease entity, only a selected number was admitted to the study.

Point 4:

Assessment methods lack an assessment of kinesiophobia and an objective evaluation of Cervical Range of Motion.

Answer 4:Thank you for your suggestion, we did not include a cervical spine mobility range test or objective tests in this study. In this work, we were particularly concerned with the subjective experiences of patients, and this was the subject of our work.

Point 5:

Please insert a flowchart of the study.

Answer 5:

Thanks for the suggestion, we've added a flowchart to the work.

Point 6: Don't you have a trial registration?

Answer 6: Our study was not registered because it is not compulsory in the country where the test is performed, however, in future studies, we will make sure that the study is registered.

Point 7:The results are short-term, the use of a follow-up is desirable, there is no blinding and no allocation. In my opinion, these criticalities should be reported in the discussion.

Answer 7: Thanks for your suggestion, we've added the limitations of the study.

Point 8: At line 125-127 “The mobilization of the cervical spine according to the concept of Orthopedic Manual Therapy by Kaltenborn and Evjenth was also used, which was conducted by qualified physiotherapy specialists.”.  What kind of mobilizations have been made? Passive? Active-assisted? Have therapeutic exercises been used beyond the stabilometric platform? Please specify that point.

Answer  8:  Thank you for the suggestion, we added the information that mobilization by traction was used. The exercises on the platform were performed in the form of an interactive game provided by the manufacturer in the device used.

Point 9: From line 286 to 294: “It has been reported that patients suffering from Degenerative Cervical Myelopathy in Neck Index Disability achieve higher scores than patients with Cervical Discopathy, which indicates that they subjectively assess their disability higher than 288 patients with Cervical Discopathy. It was also noted that patients with Degenerative Cervical Myelopathy had were more impaired in terms of concentration-related activities 290 such as reading, focusing, driving, sleeping and relaxing. In terms of pain intensity and daily activities such as care, lifting, patients with Cervical Discopathy and Degenerative Cervical Myelopathy show a similar level of discomfort associated with the disease. This is important from the perspective of the possibility of differentiating these diseases.” There are several red flags used in physiotherapy that guide the physiotherapy diagnosis and help the therapeutic action in understanding when a referral is needed. I think it should be specified that this element can be useful in screening the pathology before the diagnostic finding (MRI). It could be a guide for rehabilitators who have direct access with this type of patient.

Answer 9:

Thank you for your suggestion, we've added a clinical implications section

Round 2

Reviewer 2 Report

The authors have improved the manuscript 

minor issues remain

The randomization is still not clear for me , in my point of view there is no randomization 

There is still some tables with single row which need to be regrouped or deleted and replaced by sentences 

Author Response

Dear Reviewer,

Thank you very much for your time and valuable comments, which all have been considered and incorporated. The detailed list of responses is given below. We hope that the modifications and explanation will be acceptable for you.

Yours sincerely,

Rydzik, corresponding author

The randomization is still not clear for me , in my point of view there is no randomization 

A: This has been corrected 

There is still some tables with single row which need to be regrouped or deleted and replaced by sentences 

A: This has been corrected 

Reviewer 3 Report

Thank you for making the required changes and implementing the missing parts.

Author Response

Thank you